# Characteristics and Mechanism of Ammonia Nitrogen Removal by Heterotrophic Nitrification Bacterium *Klebsiella pneumoniae* LCU1 and Its Application in Wastewater Treatment

**DOI:** 10.3390/microorganisms13020297

**Published:** 2025-01-29

**Authors:** Meng Xu, Lifei Chen, Yizhen Xin, Xiangyu Wang, Zhuoya Wang, Xueqiang Meng, Wenyu Zhang, Haoyang Sun, Yifan Li, Wenzhe Zhang, Peng Wan, Bingshuai Geng, Lusheng Li

**Affiliations:** 1School of Agricultural Science and Biology, Liaocheng University, Liaocheng 252000, China; xu1029meng@foxmail.com (M.X.); 2210190117@stu.lcu.edu.cn (Y.X.); 2022404144@stu.lcu.edu.cn (X.W.); 2022404149@stu.lcu.edu.cn (Z.W.); 2023405028@stu.lcu.edu.cn (X.M.); 2023405034@stu.lcu.edu.cn (W.Z.); 13210751109@163.com (H.S.); 2022404118@stu.lcu.edu.cn (Y.L.); 2022404088@stu.lcu.edu.cn (W.Z.); 2022404136@stu.lcu.edu.cn (P.W.); s15098415031@163.com (B.G.); 2Shandong Province Engineering Research Center of Black Soldier Fly Breeding and Organic Waste Conversion, Liaocheng University, Liaocheng 252000, China

**Keywords:** *Klebsiella pneumoniae*, heterotrophic nitrification, wastewater treatment

## Abstract

In this study, a novel strain exhibiting heterotrophic nitrification was screened; subsequently, the strain was identified as *Klebsiella pneumoniae* LCU1 using 16S rRNA gene sequencing. The aim of the study was to investigate the effects of external factors on the NH_4_^+^-N removal efficiency of strain LCU1 in order to elucidate the optimal conditions for NH_4_^+^-N removal by the strain and improve the removal efficiency. The findings indicated that the NH_4_^+^-N removal efficiency of the strain exceeded 80% under optimal conditions (sodium succinate carbon source, C/N ratio of 10, initial pH of 8.0, temperature of 30 °C, and speed of 180 rpm). The genome analysis of strain LCU1 showed that key genes involved in nitrogen metabolism, including *nar*GHI, *nir*B, *nxr*AB, and *nas*AB, were successfully annotated; *hao* and *amo* were absent, but the nitrogen properties analysis determined that the strain had a heterotrophic nitrification ability. After 120 h, the NH_4_^+^-N removal efficiency of strain LCU1 was 34.5% at a high NH_4_^+^-N concentration of 2000 mg/L. More importantly, the NH_4_^+^-N removal efficiency of this strain was above 34.13% at higher Cu^2+^, Mn^2+^, and Zn^2+^ ion concentrations. Furthermore, strain LCU1 had the highest NH_4_^+^-N removal efficiency of 34.51% for unsterilised (LCU1-OC) aquaculture wastewater. This suggests that with intensive colonisation treatment, the strain has promising application potential in real wastewater treatment.

## 1. Introduction

Excess nitrogen accumulation is a major environmental challenge and is associated with increasing agricultural activity. Wastewater from livestock farms contains abundant nitrogen in the form of ammonia nitrogen (NH_4_^+^-N); this is due to the metabolism of the proteins in the feed [1]. The Bulletin of the Second Chinese Census of Pollution Sources indicates that the emissions of pollutants in the aquatic animal and plant farming environments comprise 22,300 t of NH_4_^+^-N and 99,100 t of total nitrogen, which account for 2.3% and 3.2% of China’s overall water pollution discharges, respectively. The unutilised feed nitrogen is excreted into the water as ammonia, faeces, and residual feeds, causing nitrogen pollution [2]. The principal methods that are currently used for nitrogen removal include physicochemical (e.g., ion exchange, reverse osmosis, and breakpoint chlorination) [3] and biological nitrogen removal (BNR) processes. Owing to its advantages in terms of high efficiency and low secondary pollution, BNR is more common [4].

In recent years, the research on BNR has mainly focused on improving the efficiency of NH_4_^+^-N removal, reducing the amount of chemicals, and reducing operating costs. Researchers have seen the discovery of new nitrogen conversion pathways and new denitrification microorganisms as a breakthrough point; to date, the identified heterotrophic nitrifying bacteria (HNB) include *Pseudomonas putida* [5], *Acinetobacter* sp. [6], *Achromobacter xylosoxidans* [7], *Alcaligenes faecalis* [8], and *Zobellella taiwanensis* [9]. Compared with autotrophic nitrifying bacteria, HNB can grow and nitrify in the presence and absence of carbon sources simultaneously [10], and they can also utilise both inorganic and organic nitrogen. They require low dissolved oxygen for growth and can tolerate acidic environments; these are important assets that make HNB valuable and advantageous for applications [11]. In addition, HNB can participate more extensively in the cyclic metabolism of phosphorus and sulphur; *Pseudomonas stutzeri*, which is a typical HNB, also functions as a phosphorus-polymerising bacteria [12]. HNB has a very good application effect in real wastewater treatment; the removal of NH_4_^+^-N by *Alcaligenes faecalis* strain No.4 is 5–40 times higher than that of microorganisms under the same conditions [13]. In studies of the simultaneous degradation of organic matter in wastewater by HNB, it was found that heterotrophic nitrification could explain the achievement of complete nitrification in systems with low dissolved oxygen and high organic loads [14]. The nitrogen metabolism pathway of HNB has been thoroughly studied, as has the general formula of the nitrogen metabolism of HNB. One pathway is inorganic nitrogen nitrification metabolism (NH_4_^+^-N→NH_2_OH→NO_2_^−^-N→NO_3_^−^-N), and the other pathway is organic nitrogen nitrification metabolism (RNH_2_→RNHOH→R-NO→RNO_2_→NO_3_^−^-N) [15]. Heterotrophic nitrification and the denitrification process of each step of the reaction are completed under the actions of specific enzymes; HNB is involved in the metabolism of the substrate. First, the ammonia monooxygenase (AMO) is catalysed by ammonia nitrogen and oxidised to NH_2_OH; then, hydroxylamine oxidase (HAO) is catalysed by N_2_O; N_2_O, in the form of nitrite reductase (NIR), is converted to NO; nitrate reductase (NAR) converts NO_3_^−^-N to NO_2_^−^-N; nitric oxide reductase (NOR) converts NO to N_2_O; nitrous oxide reductase (NOS) converts N_2_O to N_2_ [16]. The *nor*, *nis*, and *nas* were successfully annotated in the whole genome of *Klebsiella pneumoniae* strain EGD-HP19-C; although it was shown to be heterotrophically nitrifying, the nitrogen metabolism pathway was not fully described [17]. *Klebsiella pneumoniae* is an HNB strain. A *Klebsiella pneumoniae* strain isolated from tylosin fermentation residues showed efficiencies of 95.31% and 83.26% in the removal of tylosin and NH_4_^+^-N, respectively [18]. *Klebsiella pneumoniae* is widespread in nature. Fang et al. screened seven strains of *Klebsiella pneumoniae* in sludge; these strains also proved to be heterotrophically nitrifying, and the NH_4_^+^-N removal rate could be as high as 97.38% [19]. However, the external factors that impede the nitrogen removal efficiency of *Klebsiella pneumoniae* remain largely uninvestigated, particularly the influence of salinity and heavy metals. Aquaculture wastewater contains high concentrations of salts and heavy metals [20], underscoring the need for further investigation into the effects of these factors on NH_4_^+^-N removal.

In this study, a *Klebsiella pneumoniae* strain was isolated from chicken manure samples using a screening process. The effects of temperature, carbon source, carbon (C)/nitrogen (N) ratio, initial pH, salinity, dissolved oxygen, initial NH_4_^+^-N concentration, and heavy metals on the NH_4_^+^-N removal efficiency of strain LCU1 were investigated to elucidate the optimal conditions for NH_4_^+^-N removal by the strain. Moreover, strain LCU1 was used in a laboratory-scale perch water treatment system to improve the removal efficiency and to ascertain its potential for aquaculture wastewater treatment.

## 2. Materials and Methods

### 2.1. Medium

The following substances were used in the study: Luria Bertani medium (LB) (g/L): 25.0 LB medium mixed powder and 1 L of distilled water; enrichment medium (g/L): 0.2 NaNO_2_, 5.0 CaCO_3_, 0.5 NaCl, 0.5 K_2_HPO_4_, 0.5 MgSO_4_·7H_2_O, 0.4 FeSO_4_·7H_2_O, and 1 L of distilled water; screening isolation medium (VM) (g/L): 2.0 acetamide, 8.2 KH_2_PO_4_, 1.6 NaOH, 0.5 MgSO_4_·7H_2_O, 0.5 KCl, 0.0005 CaSO_4_·2H_2_O, 0.0005 CuSO_4_·5H_2_O, 0.0005 FeCl_3_·6H_2_O, 0.0005 ZnSO_4_·H_2_O, and 1 L of distilled water; nitrification medium (BM) (g/L): 5.6 disodium succinate, 1.5 KH_2_PO_4_, 0.47 (NH_4_)_2_SO_4_, 7.9 Na_2_HPO_4_·7H_2_O, 0.1 MgSO_4_·7H_2_O, 2 mL of trace element solution, and 1 L of distilled water; trace element solution (g/L): 50.0 Na_2_EDTA, 2.2 ZnSO_4_·7H_2_O, 5.5 CaCl_2_, 5.06 MnCl_2_·4H_2_O, 5.0 FeSO_4_, 1.57 CuSO_4_·5H_2_O, 1.60 CoCl_2_·6H_2_O, and 1 L of distilled water; denitrification media (g/L): 0.72 KNO_3_ (ADM1) or 0.49 NaNO_2_ (ADM2), 5.62 disodium succinate, 7.9 Na_2_HPO_4_·7H_2_O, 1.5 KH_2_PO_4_, 0.1 MgSO_4_·7H_2_O, 2 mL of trace element solution, and 1 L of distilled water [21]. All the above reagents were purchased from Macklin Reagent (Shanghai, China).

### 2.2. Strain Screening, Isolation, and Identification

Chicken faecal samples were obtained from Woneng Agricultural Technology Co. Liaocheng City, Shandong Province, China, and returned to the laboratory at 4 °C. Following thorough mixing, the faecal samples (10 g) were inoculated in 300 mL triangular conical flasks containing 90 mL of sterilised enrichment medium, shaken thoroughly, and reincubated at 180 rpm at 30 °C. Enrichment was conducted for 72 h, with ammonium sulphate 28 supplemented every 24 h to eliminate the microorganisms that could not utilise NH_4_^+^-N. The enriched bacterial suspension was diluted to 10^−7^ in a gradient, and 0.1 mL of a bacterial solution diluted to 10^−5^, 10^−6^, and 10^−7^ was applied to the plate screening medium and incubated in an inverted incubator for 2~3 days at 30 °C.

Single colonies of different sizes and morphologies were picked from the primary sieving plate for isolation and purification, and the above operation was repeated three times. Single colonies were picked from the purified plates and incubated in 300 mL triangular flasks containing 100 mL of NH_4_^+^-N at 180 rpm at 30 °C for 32 h. Subsequently, the NH_4_^+^-N concentration was determined, and the strains with higher NH_4_^+^-N degradation efficiency were finally selected as the target strains. The isolates with the highest NH_4_^+^-N removal activity were kept in slants of LB agar medium at 4 °C.

Bacterial DNA extraction kits (TIANGEN, Beijing, China) were used. The 16S rDNA gene of the isolate was extracted, amplified using PCR with the bacterial universal primers 27F (5′-AGA GTT TGA TCA TGG CTC AG-3′) and 1492R (5′-TAC GGT -TAC CTT GTT ACG ACTT-3′) [3], and sequenced by TsingKe Biological Technology Co. (Qingdao, China). The amplified sequences were purified, sequenced, and compared with the 16S rRNA gene sequences of other microorganisms in the National Center for Biotechnology Information database using the BLAST programme. The phylogenetic tree was constructed via the neighbour-joining method using MEGA 11.0 software.

### 2.3. Heterotrophic Nitrification and Growth Characteristics of Strain LCU1

Before the experiment, the *Klebsiella pneumoniae* strain was activated in an LB medium until the optical density at 600 nm (OD_600_) reached 1, and 1 mL of the activated bacterial solution was inoculated into a 300 mL flask containing 100 mL of BM. The *Klebsiella pneumoniae* strain was incubated at 30 °C, with an initial pH of 7.0, at 180 rpm for 24 h. Samples were taken every 3 h to determine the OD_600_. Subsequently, the ADM1 and ADM2 media were used, and LCU1 was introduced into the media with the same inoculation and culture conditions; samples were taken every 3 h. The samples were centrifuged at 9600× *g* for 1 min, and the concentrations of N-containing compounds (NH_4_^+^-N, NO_2_^−^-N, and NO_3_^−^-N) in the supernatant were determined.

### 2.4. Heterotrophic Nitrification by Strain LCU1 Under Different Unifactorial Conditions

The heterotrophic nitrification characteristics of the strains were studied under different culturing conditions, such as carbon source, C/N ratio, initial pH, temperature, salt concentration, and heavy metal concentration. To determine the optimum carbon source for the strain, sucrose, glucose, glycerol, sodium citrate, and sodium acetate were used as the sole carbon sources for BM, replacing sodium succinate. The six carbon sources selected in this study included common organic carbon sources, such as sugars, organic acids, and small molecular organic alcohols. After determining the optimum carbon source, the effects of C/N ratios of 2, 5, 10, 15, and 20 on NH_4_^+^-N removal were investigated by varying the concentration of the optimum carbon source in the substrate. To determine the effect of strain incubation temperature on NH_4_^+^-N removal, the shaker was set to 20, 25, 30, 35, and 40 °C. To clarify the effect of the initial pH on the efficiency of NH_4_^+^-N removal by the strain, the initial pH of the BM medium was adjusted to 5, 6, 7, 8, 9, and 10 using 0.1 mol/L NaOH and 0.1 mol/L HCl. The effect of salinity on the NH_4_^+^-N removal capacity of the strain was investigated by adjusting the salinity through the addition of NaCl to the BM at final concentrations of 0, 10, 20, 30, 40, and 50 g/L. The effect of heavy metals on the removal of NH_4_^+^-N was investigated using Cu^2+^, Zn^2+^, and Mn^2+^ as heavy metal ions at final concentrations of 5, 10, and 20 mg/L, respectively. The samples were centrifuged at 9600× *g* for 1 min, and the concentrations of N-containing compounds (NH_4_^+^-N, NO_2_^−^-N, and NO_3_^−^-N) in the supernatant were determined.

### 2.5. Removal of High Concentrations of NH_4_^+^-N

To investigate the efficiency of bacteria in removing high NH_4_^+^-N concentrations, different amounts of ammonium sulphate were added to 100 mL of BM filtered through a membrane filter with a pore size of 0.22 μm to final concentrations of 500, 1000, 1500, and 2000 mg/L NH_4_^+^-N. Similarly, 1 mL of the log-phase bacterial broth was added to the BM media with different initial NH_4_^+^-N concentrations, and the medium was incubated under optimal conditions for 120 h. The NH_4_^+^-N concentration and OD_600_ were measured periodically during the incubation.

### 2.6. Genome Sequencing and Gene Annotation

Whole genome sequencing of strain LCU1 was conducted using a combination of whole genome sequencing technology from Majorbio (Shanghai, China) and Illumina and PacBio sequencing technologies. All of the analyses were performed using the online platform of Majorbio Cloud Platform (http://cloud.majorbio.com accessed on 20 May 2024) from Shanghai Majorbio Bio-pharm Technology Co., Ltd. The resulting sequencing data exhibited a coverage of 124×, and 72,040 high-quality reads were reassembled using the SOAPdenovo (2.04) genomics workbench. Glimmer was employed for the purpose of CDS prediction, with the relevant annotations being obtained from the NR, Swiss-Prot, and KEGG databases. This was achieved by means of the utilisation of sequence alignment tools such as BLAST, Diamond, and HMMER. In summary, each set of query proteins was aligned with the aforementioned databases, and annotations of the best-matched subjects (e-value < 10^−5^) were obtained for gene annotation.

### 2.7. Aquaculture Wastewater Treatment with Strain LCU1

Real aquaculture wastewater from the Liaocheng Wan Shun fishing farm was collected to explore the application potential of strain LCU1. Strain LCU1 was pre-cultured in LB to the logarithmic growth stage. In the autoclave group, 100 mL of wastewater culture was pre-autoclaved at 121 °C for 15 min, while the non-autoclaved group was not subjected to autoclaving. Similarly, 1% of the pre-culture strain solution was directly added to the LCU1 group; sodium succinate was added to adjust the carbon/nitrogen ratio (C/N ratio = 10), and 1% of the bacterial solution was added to the bio-enhanced treatment group (LCU1-OC group). The wastewater without treatment was used as the blank group. All the treatments were conducted three times and shaken at 180× *g* at 30 °C for 24 h. The initial and final concentrations of NH_4_^+^-N were determined.

### 2.8. Analytical Methods

Bacterial growth (OD_600_) was determined at 600 nm using a spectrophotometer (Multiskan Sky, Thermo, Shanghai, China). The concentrations of NH_4_^+^-N, NO_2_^−^-N, and NO_3_^−^-N were determined using standard methods [22]. The NH_4_^+^-N removal efficiency formula was (C_1_-C_n_)*100%/C_1_, where C1 was the initial concentration, and C_n_ was the final concentration at 24 h. The results were expressed as the standard deviation of three replicates. A one-way analysis of variance was performed using SPSS statistical software 17.0 and Tukey’s HSD test (*p* < 0.05), and Graphpad software (Version 9.0.0) and Microsoft Excel were used for graphical processing.

## 3. Results

### 3.1. Isolation and Identification of the Klebsiella pneumoniae Strain LCU1

Strain LCU1 with good nitrogen metabolism ability was screened for further analysis. The biochemical and physiological characteristics showed that strain LCU1 was a Gram-negative bacterium (Figure 1A). The mature colonies were rounded, greyish white with wrinkled edges that protruded in the centre, pale white, translucent, moist, and sticky, and they could be easily pulled into filaments using inoculation loops on an LB solid agar medium (Table 1). A comparison of the physiological and biochemical characteristics of strain LCU1 with those described in Bergey’s Manual of Systematic Bacteriology revealed that strain LCU1 exhibited the defining characteristics of the *Klebsiella* genus. The strain image was captured using a 50k× electron microscope (low-vacuum SEM, JSM-6390LV, Japan). The bacterial cells were observed under a scanning electron microscope and exhibited a rod-like morphology, with a length of approximately 1.0–2.0 μm and a width of 0.6–1.0 μm (Figure 1B). The cells displayed a rod-like configuration (Figure 1B). The sequence homology comparison revealed that strain LCU1 was closely related to the *Klebsiella pneumoniae* strain DSM 30104, with 99.86% 16S ribosomal RNA similarity. Based on partial 16S rDNA sequences, a phylogenetic tree was constructed in MEGA 11.0 with adjacent relationships (Figure 1C).

### 3.2. Growth and Heterotrophic Nitrification Characteristics of Strain LCU1

Under aerobic conditions, the heterotrophic nitrification capacity and cell growth of strain LCU1 were demonstrated in a nitrification medium with NH_4_^+^-N as the sole nitrogen source (Figure 2). Strain LCU1 reached its maximum growth between 9 and 12 h and subsequently entered the stationary phase after 21 h of incubation. The time course of OD_600_, NH_4_^+^-N, NO_2_^−^-N, NO_3_^−^-N, and LCU1 growth during nitrification under optimal conditions was studied (Figure 2). Concerning cell growth, the biomass of strain LCU1 increased rapidly between 6 and 18 h, corresponding to an increase in OD_600_ from 0.0640 to 0.8471. However, the biomass of strain LCU1 exhibited no significant changes after 21 h. Ammonium nitrogen (with an initial concentration of 100 mg/L) exhibited a markedly decreased concentration within 12 h and was completely removed within approximately 24 h. LCU1 had the highest NH_4_^+^-N removal capacity at 9–12 h and reached 16.3986 mg/L/h. NO_3_^−^-N accumulation was observed, reaching 95.8632 mg/L of NO_3_^−^-N at 24 h. The process did not result in NO_2_^−^-N accumulation, with the concentration remaining at a low level (<0.5 mg/L). The growth efficiency and OD_600_ of strain LCU1 in the ADM1 and ADM2 media were lower than those of the media with (NH_4_)_2_SO_4_ as the nitrogen source (Figure 3A,B), indicating that strain LCU1 could not utilise nitrite under aerobic conditions.

### 3.3. Heterotrophic Nitrification Under Various Conditions

#### 3.3.1. Effects of Carbon Source on NH_4_^+^-N Removal

The carbon source is usually considered a significant factor influencing nitrogen removal because it provides an energy source for heterotrophic bacteria. In this study, the effects of different carbon sources, such as sodium succinate, sucrose, glucose, glycerol, sodium acetate, and sodium citrate, on nitrification capacity and cell growth were investigated (Figure 4A). Significant differences were observed using different carbon sources. Among the carbon sources, the nitrification efficiency of the cells was relatively high when sodium citrate and sodium succinate were used as the carbon sources, with NH_4_^+^-N removal efficiencies of 86.76% and 93.90%, respectively. In addition, biomass analysis showed that when sodium citrate and sodium citrate were added as the sole carbon sources, the OD_600_ reached 0.82 and 0.87, respectively. Sodium citrate and sodium succinate are used as carbon sources for most heterotrophic nitrifying bacteria. Therefore, in this study, sodium succinate was selected as the carbon source for nitrification performance in the subsequent analyses.

#### 3.3.2. Effect of C/N Ratio on NH_4_^+^-N Removal

The effects of varying C/N ratios on cell growth and the NH_4_^+^-N removal capacity of LCU1 in BM were examined by altering the sodium succinate concentration (Figure 4B). Regarding bacterial growth, the biomass exhibited an increase with increasing C/N ratios, reaching a maximum (OD_600_ = 0.83) at a C/N ratio of 10:1. For the efficiency of NH_4_^+^-N removal, the C/N ratio gradient exhibited significant variation, spanning a range from 2:1 to 20:1. The removal of NH_4_^+^-N increased with an increasing C/N ratio, reaching a maximum at a C/N ratio of 10; subsequently, the removal efficiency decreased. The lower degradation of NH_4_^+^-N at C/N ratios of 2:1 and 5:1 may be attributed to an insufficient carbon source and the rapid compensatory depletion of succinic acid.

#### 3.3.3. Effect of Temperature on NH_4_^+^-N Removal

The effects of temperature on biomass and the NH_4_^+^-N removal characteristics of *Klebsiella* sp. were determined. LCU1 (Figure 4C) growth and nitrogen removal efficiency were minimal at incubation temperatures of 10 and 37 °C. The optimum removal efficiency of strain LCU1 was observed at 30 °C, with a maximum removal efficiency of 86.42% and an OD_600_ maximum of 0.76. In addition, the strain maintained a strong nitrification efficiency at 40 °C.

#### 3.3.4. Effect of Initial pH on NH_4_^+^-N Removal

The cell growth and NH_4_^+^-N removal efficiency of LCU1 were determined at different pH values (Figure 4D). Strain LCU1 showed efficient nitrification characteristics in the initial pH range of 5~10. Strain LCU1 grew optimally over a wide pH range and showed efficient nitrogen removal ability. The highest removal efficiency of 87.77%, with an OD_600_ value of 0.86, was achieved during incubation at pH 7 for 24 h under initial conditions; however, strain LCU1 was affected by strongly acidic conditions, and its growth experienced marked inhibition under an alkaline condition.

#### 3.3.5. Effect of Dissolved Oxygen on NH_4_^+^-N Removal

Heterotrophic nitrifying strains are mostly aerobic; therefore, different dissolved oxygen concentrations markedly affect the growth, nitrification process, and nitrification products of the strains, and the strains are even more affected by oxygen concentration. The effects of rotational speed on the growth and NH_4_^+^-N removal characteristics of strain LCU1 were determined (Figure 4E). At rotational speeds of 90, 120, 150, 180, and 210 rpm, strain LCU1 grew rapidly and reached the maximum OD_600_ value at 12 h, and its OD_600_ values at the different rotational speeds were 0.47, 0.58, 0.77, 0.91, and 0.81, respectively. The NH_4_^+^-N removal efficiencies of strain LCU1 were relatively high under these conditions and were 41.44%, 58.53%, 80.76%, 90.21%, and 86.82%, respectively. Among the speeds, 180 rpm was the most suitable for strain growth and was associated with the highest NH_4_^+^-N removal efficiency.

#### 3.3.6. Effect of Salinity on NH_4_^+^-N Removal

Salinity represents a significant factor influencing the removal of NH_4_^+^-N and the growth of heterotrophic nitrifying bacteria. There was a notable decline in NH_4_^+^-N removal and the OD_600_ values with increasing salinity (Figure 4F). At salinities below 30 g/L NaCl, strain LCU1 demonstrated a high tolerance level with a high NH_4_^+^-N removal efficiency. This indicates that strain LCU1 is a salinophilic bacterium that thrives in high-salinity environments. The removal efficiency was 38.4%, and the OD_600_ value was 0.37 at 50 g/L NaCl. High salinity-induced high osmolarity reportedly causes cytoplasmic lysis and cellular or enzymatic activity loss.

#### 3.3.7. Effect of Heavy Metals on NH_4_^+^-N Removal

Heavy metals affected the cell growth and NH_4_^+^-N removal efficiency of strain LCU1. Strain LCU1 showed varying degrees of resistance to some of the heavy metals (Figure 4G). Cu^2+^ inhibited strain LCU1 more effectively than Zn^2+^ and Mn^2+^. Strain LCU1 was more sensitive to Cu^2+^, and the NH_4_^+^-N removal efficiency was 50.7% at 10 mg/L Cu^2+^. The NH_4_^+^-N removal efficiencies were 59.1% and 58.8% at Zn^2+^ and Mn^2+^ concentrations of 10 mg/L, respectively. Similarly, 20 mg/L of heavy metals had a strong inhibitory effect on cell growth and NH_4_^+^-N removal efficiency.

#### 3.3.8. Effect of a High Level of NH_4_^+^-N on NH_4_^+^-N Removal

The impact of elevated ammonium levels on the NH_4_^+^-N removal efficiency of strain LCU1 was examined at initial ammonium concentrations of 500, 1000, 1500, and 2000 mg/L. The NH_4_^+^-N removal efficiency of strain LCU1 under high ammonium concentrations exhibited a notable decline over the initial 120 h (Figure 4H). Subsequently, the NH_4_^+^-N removal efficiencies reached 58.95%, 52.83%, and 49.91% at initial NH_4_^+^-N concentrations of 500, 1000, and 1500 mg/L, respectively. However, at an NH_4_^+^-N concentration of 2000 mg/L, cell growth was inhibited, with an NH_4_^+^-N removal efficiency of 33.94% at 120 h. The findings indicate that the elevated ammonium concentration diminished the NH_4_^+^-N removal efficiency of strain LCU1, while the inhibitory impact intensified with the initial concentration.

### 3.4. Genomic Analysis of Strain LCU1

To gain further insights into the denitrification pathway of strain LCU1, we conducted a comprehensive genomic analysis, encompassing sequencing, assembly, and annotation of the strain’s genome. The genome size was 5,751,502 bp (including scaffolds), and the size of the chromosome genome was 5,404,767 bp. Additionally, the repeat sequence was 20,782 bp, constituting 0.41% of the total length. The entire genome of LCU1 was uploaded to the NCBI (National Center for Biotechnology Information) database (accession: PRJNA1199685) and mapped based on the sequencing (Figure 5). The LCU1 genome comprises a circular chromosome and two plasmids. The prediction analysis showed that there were 5342 coding genes in the gene sequences, with the total length of the predicted coding genes being 501,9144 bp. The gene average length of the sequence was 939.56 bp, with a GC content of 58.48%. Furthermore, there were 25 rRNAs (comprising 8 23S rRNAs, 8 16S rRNAs, and 9 5S rRNAs) and 86 tRNAs. A summary of the characteristics of the assigned genes is provided in the Appendix A. Notably, 5338 and 4728 genes were annotated through the NR and Swiss-Prot databases, respectively. The genes associated with the nitrogen metabolism pathway were studied using the KEGG (Kyoto Encyclopaedia of Genes and Genomes) database. The strain demonstrated HNB functionality based on preliminary experimental analysis; however, KAAS (KEGG Automatic Annotation Server) was able to reveal the key genes involved in nitrification.

To further substantiate the presence of HNB-related genes in the genome of strain LCU1, we annotated 40 nitrogen metabolism pathway-related genes based on the NR and Swiss-Prot databases. Strain LCU1 transported NO_3_^−^-N into the cell in the presence of *nrt* and reduced NO_3_^−^-N to NO_2_^−^-N, when catalysed by nitrate reductase *nar* GHI. NO_2_^−^-N was reduced to N_2_O by nitrite reductase (*nir*S); however, nitrous oxide reductase, which reduces the resulting N_2_O to NO, was not detected. *amo*A and *amo*B were absent, indicating the potential existence of a novel *amo* in strain LCU1.

Therefore, it is hypothesised that strain LCU1 performs HNB via the newly identified mechanism (Figure 6). The genome of strain LCU1 contains various functional genes involved in different nitrogen metabolism pathways. This is the inaugural report on the coexistence of these functional genes (*nar*GHI, *nir*BD, *nas*AB, *nir*A, and *nif*D) in a single HNB strain. Based on the functional genes identified, multiple nitrogen metabolic pathways were predicted, including assimilatory nitrate reduction and ammonium assimilation; via the hydroxylamine pathway, HNB utilised unidentified enzymes that were functionally analogous to the Amo and Hao enzymes. In addition, we labelled the genes of the glutamine synthetase–glutamate synthase pathway (*gln*A), which are involved in the intracellular ammonium assimilation process. Similarly, we observed that strain LCU1 had nitrogen fixation genes (*nif*) that were responsible for manipulating and regulating the synthesis of nitrogenase (nitrogen fixing enzyme). Genomic analysis was conducted to identify the genes closely related to the nitrification process. This analysis revealed the presence of genes encoding key nitrification enzymes, such as ammonia monooxygenase and hydroxylamine oxidase. The findings of this study suggest that strain LCU1 is capable of nitrification.

### 3.5. Strain LCU1 Application in Real Aquaculture Wastewater Treatment

In this study, the LCU1 group showed low NH_4_^+^-N removal efficiency (<10%), which was not significantly different from that of the blank group (Figure 7). This was primarily due to the low concentration of organic matter in aquaculture wastewater, which could not provide sufficient effective organic carbon for the bacteria. The NH_4_^+^-N concentration in the LCU1-OC group was significantly reduced during aquaculture wastewater treatment. Considering that the presence of antibiotic residues in the wastewater limited the growth of the unsterilised group and that high-temperature sterilisation led to the denaturation and inactivation of the antibiotics, the addition of LCU1 improved the NH_4_^+^-N removal efficiency in the sterilised and unsterilised groups compared with the blank control group. After adjusting the C/N ratios, the NH_4_^+^-N removal efficiency was significantly increased, particularly in the sterilised group, which reached 83.70%. The nitrogen removal effect of the sterilised group was better than that of the non-sterilised group. The above results indicate that strain LCU1 aids in treating aquaculture wastewater through bioaugmentation with the addition of an organic carbon source (C/N ratio = 10).

## 4. Discussion

Most studies on the NH_4_^+^-N removal ability of *Klebsiella* sp. have focused on its NH_4_^+^-N removal effect, with minimal investigation into its removal mechanism and practical application in low-carbon aquaculture wastewater. Strain LCU1, as an HNB, has the advantages of a fast growth rate, strong resistance to environmental stress, a simplified process, and high NH_4_^+^-N removal efficiency. Strain LCU1, in the ideal state, has an ammonium nitrogen removal efficiency of up to 99.26%, compared with *Klebsiella* sp. TN-10, which has an efficiency of 96% [23]. In addition to its excellent NH_4_^+^-N removal ability, it was observed that strain LCU1 achieved a maximum NH_4_^+^-N removal efficiency of 16.40 mg/L/h in optimum conditions, which was faster than that of *Acinetobacter* sp. T1, which exhibited an NH_4_^+^-N removal efficiency of 12.08 mg/L/h [24].

The investigation of the optimal conditions for NH_4_^+^-N removal by strain LCU1 showed that the carbon source significantly influences the cell growth and NH_4_^+^-N removal efficiency of the strain. Strain LCU1 more readily utilises small-molecule carbon sources, such as sodium succinate and sodium citrate, in a similar manner to *Fusobacterium johannes* strain ZHL01, which also has a higher NH_4_^+^-N removal efficiency when utilising small-molecule carbon sources. ZHL01 tends to metabolise the organic matter using the tricarboxylic acid cycle when it utilises sodium fumarate or sodium succinate as a carbon source. Hence, when strain LCU1 utilises sodium succinate, the compound is directly involved in the metabolic process and provides energy rapidly, resulting in a better NH_4_^+^-N removal performance [25]. Similarly, the C/N ratio significantly affects the growth of heterotrophic bacteria. Under low C/N conditions, cell growth is limited by carbon, and the limitation of cell growth and ammonia oxidation by the shortage of energy and electron donors leads to a relatively low NH_4_^+^-N removal efficiency; therefore, LCU1 shows a low NH_4_^+^-N removal efficiency at a C/N ratio of ≤5. However, an excessively high C/N ratio causes insufficient nitrogen supply, severely limiting cell growth and metabolism and leading to a reduced NH_4_^+^-N removal efficiency. The appropriate C/N ratio is essential to maintain the normal functioning of microbial cells, while the ratio of carbon to nitrogen ratio also affects the growth of the strain and the ammonia nitrogen removal efficiency. Therefore, the effect of the C/N ratio on strain LCU1 is consistent with this inference, and its optimal C/N ratio is 10. In studies of the optimal C/N ratio, other nitrifying bacteria strains showed similar results, such as *Pseudomonas stutzeri* YG-24 with an optimal C/N ratio of 8:1 [26] and *Agrobacterium* sp. LAD9 with a C/N ratio of 8:1 [27]. The NH_4_^+^-N removal efficiency of strain LCU1 was affected by different temperature conditions. An increased temperature resulted in an initial increase in NH_4_^+^-N removal, which subsequently decreased. This suggests that high temperatures may inactivate enzymes, nucleic acids, and other cellular components, leading to metabolic blockage and bacterial death. However, within the range of temperatures tolerated by the enzyme activity, higher temperatures resulted in a faster enzyme reaction rate [28]. Most HNB strains, such as *Pseudomonas stutzeri* C3, are reportedly mesophilic and exhibit optimal growth at temperatures between 30 °C and 35 °C [29]. An investigation of the initial pH showed that strain LCU1 adapted to a pH of 5–10 and maintained a > 60% efficiency for removing NH_4_^+^-N in the pH range of 6–8. A slightly alkaline environment benefits the nitrifying bacteria because the medium contains more free ammonium, and the AMO utilises NH_3_ as the actual substrate rather than preferentially utilising NH_4_^+^-N [30]. As a crucial factor affecting microorganism growth and metabolism, the DO (dissolved oxygen) concentration markedly affects the growth and metabolism of aerobic or anaerobic microorganisms, and the high environmental DO concentration favours aerobic microorganisms. Nitrifying bacteria are mostly aerobic, and aerobic microorganisms use oxygen as an electron acceptor for respiration and to obtain energy [31]. Previous studies have also shown that the degradation efficiency of ammonium nitrogen by KSND decreased with a decreased DO content; thus, the moderate DO concentration could provide a better environment for nitrification and denitrification for the energy metabolism of the cells of strain LCU1, improving its NH_4_^+^-N removal efficiency [32].

Furthermore, salinity is a significant factor influencing the growth and NH_4_^+^-N removal efficiency of heterotrophic nitrifying bacteria. Strain LCU1 demonstrated the capacity to thrive and remove over 50% of NH_4_^+^-N within a specific salinity range (0–20 mg/L). An excessively high salinity inhibited the growth of the strain and reduced its NH_4_^+^-N removal ability. The results were similar to those previously reported for other strains, such as *Pseudomonas* sp. ADN-42 [33]. This could be because the salinity level affects the balance of the microbial intracellular osmotic pressure. In high-salinity environments, microbial intracellular water infiltrates the extracellular space, inhibiting microbial growth, and it may even lead to excessive water loss and cell death [34]. In practical wastewater treatment, the wastewater usually exhibits high salinity; hence, assessing the tolerance of the selected strains to high salinity levels is essential to ascertain their suitability for application in wastewater treatment.

Heavy metal ions are a common pollutant in wastewater; therefore, this study investigated the effects of different heavy metal ions and their varying concentrations on NH_4_^+^-N removal efficiency. As the concentration of the three heavy metal ions increased, the normal metabolic activity of strain LCU1 was disrupted. Despite a notable reduction in its NH_4_^+^-N removal efficiency, it exhibited a higher efficiency than *Aeromonas* sp. HN-02 [35] and *Alcaligenes faecalis* SDU20 [21] at 20 mg/L. This suggests that strain LCU1 is a promising treatment agent for aquaculture wastewater containing high concentrations of heavy metal ions. In the actual treatment of aquaculture wastewater, high initial NH_4_^+^-N content in wastewater is a common challenge. High NH_4_^+^-N affects the osmotic pressure of the cells and has a toxic effect on cell growth. Furthermore, it activates the substrate inhibition effect, which restricts the activity of the crucial enzymes in the nitrification reaction of the strains and, thus, affects the nitrification activity of the nitrifying bacteria [9]. This indicates that as the initial NH_4_^+^-N concentration is increased, the NH_4_^+^-N removal efficiency of LCU1 decreases. However, notably, at an initial NH_4_^+^-N concentration of 2000 mg/L, the removal efficiency remained at 33.94%, suggesting that LCU1 exhibits a higher tolerance than *Acinetobacter junii* YB; however, this requires further investigation in practical wastewater treatment [36].

Regarding the annotation of the heterotrophic nitrification efficiency and nitrogen metabolic pathway of strain LCU1, this study showed that the strain could not directly utilise NO_3_^−^-N and NO_2_^−^-N when KNO_3_ and NANO_2_ were used as the sole nitrogen sources. However, strain LCU1 exhibited normal growth. The annotation results revealed the presence of *nif* in strain LCU1, which was postulated to facilitate nitrogen conversion by LCU1 through the action of nitrogen fixation enzymes. The same nitrogen fixation genes were detected in the genus *Klebsiella michiganensis* [37]. The key genes in the metabolism of heterotrophic nitrifying bacteria are *amo* and *hao*, but neither of these genes was annotated in LCU1; thus, it was speculated that genes with similar effects may exist [38]. When NH_4_^+^-N was the only nitrogen source, NO_3_^−^-N accumulation was detected, and it was speculated that NH_4_^+^-N was converted to hydroxylamine in the process and that hydroxylamine was converted to NO_2_^−^-N through a pathway and was oxidised to NO_3_^−^-N by NXR, causing NO_3_^−^-N accumulation. This nitrification process was rapid, as no NO_2_^−^-N accumulation was detected, and a similar pathway was postulated in the study of SND-01 [39]. The unique nitrogen metabolism mechanism of LCU1 can effectively remove NH_4_^+^-N but lacks the ammonia oxidation of the traditional nitrification pathway of ammonia oxidation genes. For example, *Pseudomonas putida* Y-9 can oxidise NH_4_^+^-N to nitric oxide and can further oxidise it to nitrous oxide under aerobic conditions, with no accumulation of hydroxylamine as an intermediate in the whole process and without the presence of the *amo* in the genome of its strain [40]. Another speculation is that there may be alternatives or redundant genes of *amo* and *hao* genes in LCU1, which may have similar functions but may differ in their sequences from the known genes; therefore, they cannot be accurately identified using the existing annotation algorithms. It is also possible that the gene annotation is based on existing databases and comparison algorithms; however, for some of the newly discovered strains or strains with special metabolic pathways, the existing databases may not be able to accurately annotate all their genes.

In the aquaculture wastewater treatment, our findings suggest that the sterilised group demonstrated a higher NH_4_^+^-N removal efficiency than the unsterilised group. This hypothesis is based on the premise that there may be residual antibiotics in the aquaculture wastewater [41], potentially inhibiting bacterial growth. Following high-temperature sterilisation, the effective antibiotic components may have been inactivated, enabling strain LCU1 to grow normally. LCU1 is a *Klebsiella pneumoniae* strain isolated from chicken faeces. Hence, in the process of practical application, its direct use in the breeding environment should be avoided, and unprotected direct contact between operators and strain LCU1 should also be avoided in order to reduce its pathogenic potential. The resistance of *Klebsiella pneumoniae* to antibiotics varies among different strains, and although previous studies have shown that strain TN-1 can degrade tylosin [18], strain LCU1 has a very low ammonia nitrogen removal rate in aquaculture wastewater in which antibiotics have not been inactivated, possibly because LCU1 is sensitive to antibiotics. Strain LCU1 demonstrated effective NH_4_^+^-N removal in aquaculture wastewater treatment. However, aquaculture wastewater has a low carbon level, and the removal of its NH_4_^+^-N by LCU1 requires strengthening through the addition of external carbon sources. Furthermore, the NH_4_^+^-N removal capacity of the strain can be augmented by encapsulating the microorganisms in biopolymer microcapsules with organic substrates [42]. The co-culture of fungi and bacteria for the removal of various organic pollutants [43] is used to investigate their potential for application in wastewater treatment and to identify suitable bacterial resources for wastewater denitrification treatment.

## 5. Conclusions

A strain of HNB, designated as LCU1, was isolated from livestock effluent. The morphological characteristics and 16S rRNA gene homology analyses indicated that LCU1 is a *Klebsiella pneumoniae* strain. The NH_4_^+^-N removal efficiency of LCU1 was influenced by the carbon source, C/N ratio, temperature, pH, and shaker speed. Whole genome sequencing and gene amplification suggested that the pathway of strain LCU1 was involved in the NH_4_^+^-N removal. Strain LCU1 was effective in removing NH_4_^+^-N from farming wastewater, particularly after the adjustment of the C/N ratio for the farm wastewater.

## Figures and Tables

**Figure 1 microorganisms-13-00297-f001:**
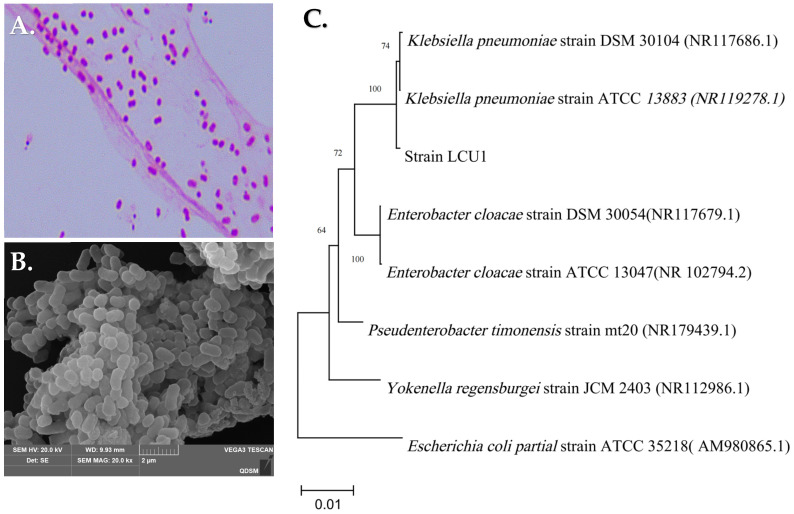
Morphological observation of strain LCU1: (**A**) Gram staining of LCU1. (**B**) Scanning electron microscope image of LCU1. (**C**) The phylogenetic tree of LCU1 derived from neighbour-joining analysis of partial 16S rRNA sequences.

**Figure 2 microorganisms-13-00297-f002:**
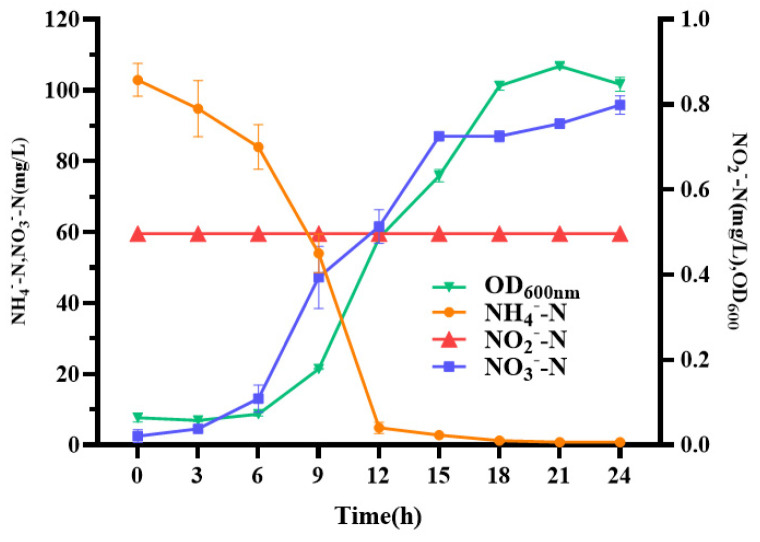
Growth and heterotrophic nitrification characteristics of strain LCU1.

**Figure 3 microorganisms-13-00297-f003:**
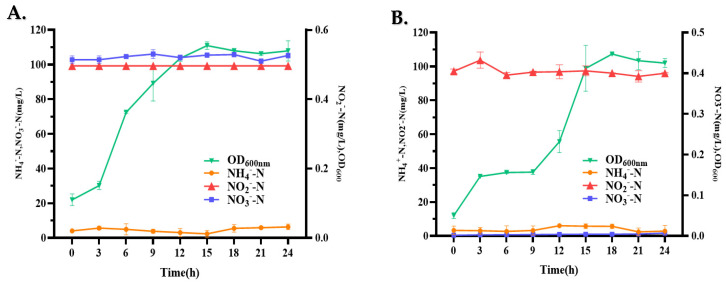
Heterotrophic nitrification characteristics of strain LCU1 growth in (**A**) KNO_3_ and (**B**) NaNO_2_.

**Figure 4 microorganisms-13-00297-f004:**
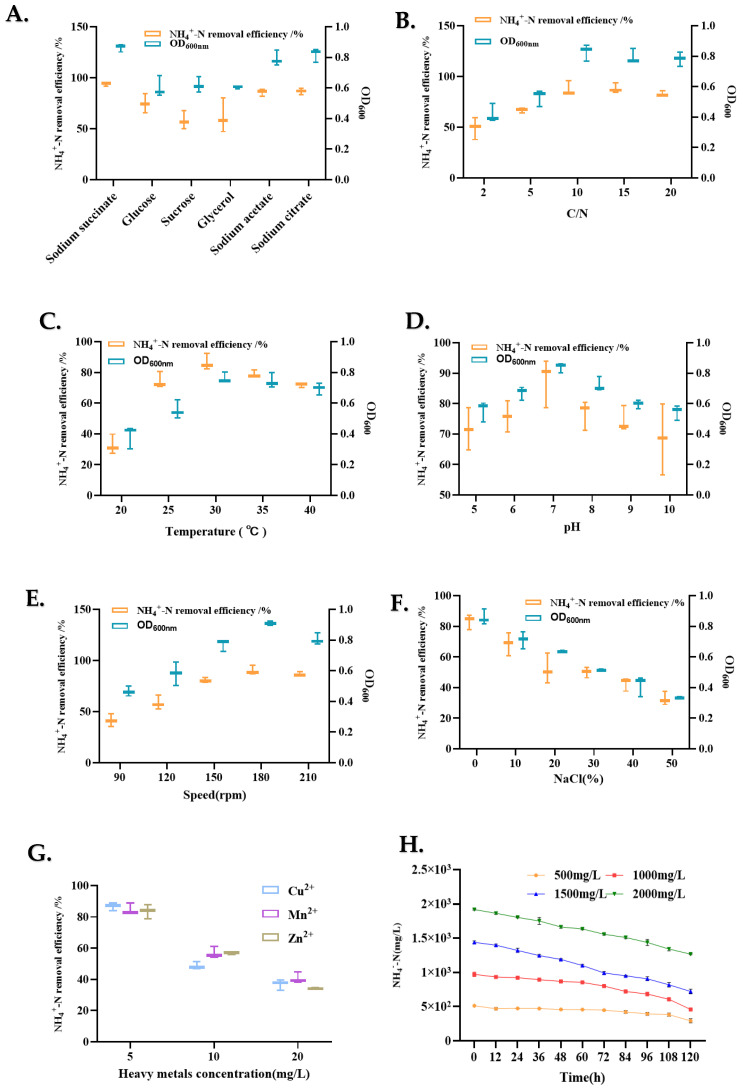
Effect of single factor on NH_4_^+^-N removal and cell growth of strain LCU1. OD_600_ value data were obtained after strain LCU1 had been cultured for 24 h: (**A**) carbon source, (**B**) C/N ratio, (**C**) temperature, (**D**) initial pH, (**E**) speed, (**F**) salinity, (**G**) heavy metals, and (**H**) initial high NH_4_^+^-N concentration.

**Figure 5 microorganisms-13-00297-f005:**
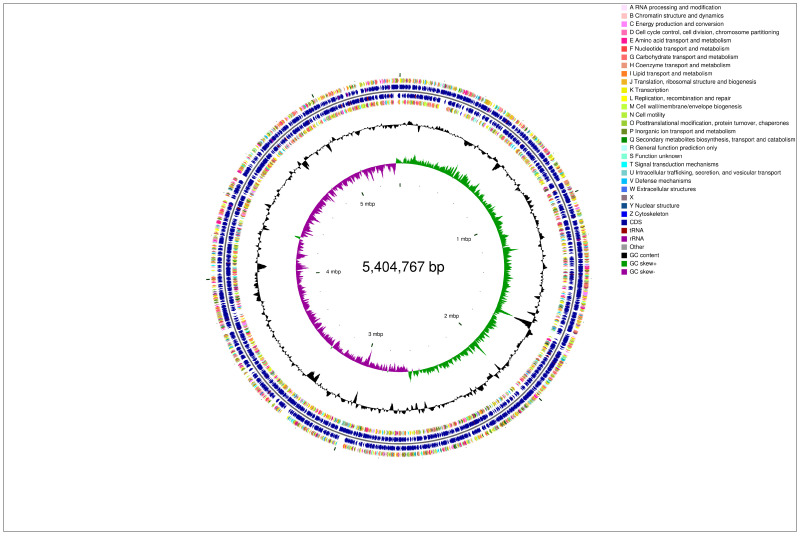
Circular graphic visualisation of the distribution of strain LCU1 genome annotations. Genome wide circle mapping of LCU1 genome circle layers from inside to outside: genome size identifier, coding sequence and COG functional classification, rRNAs and tRNAs, GC content, and GC values.

**Figure 6 microorganisms-13-00297-f006:**
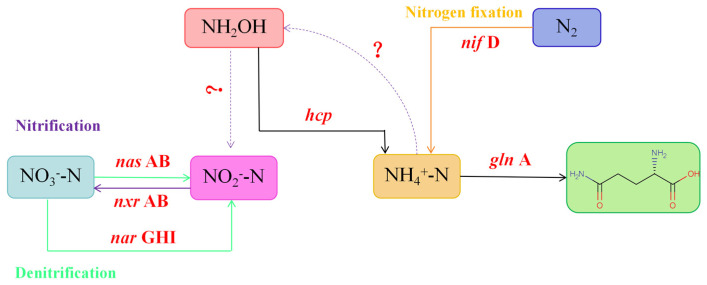
The key genes of LCU1 in the nitrogen metabolism pathway were successfully annotated in the KEGG database (green signifies the denitrification pathway; purple denotes the nitrification pathway; orange represents the nitrogen fixation pathway; question marks are used to indicate that the relevant genes were not annotated).

**Figure 7 microorganisms-13-00297-f007:**
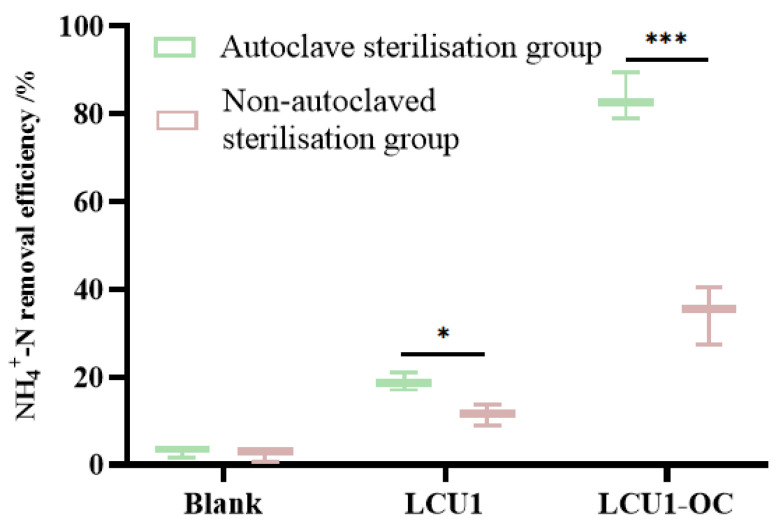
NH_4_^+^-N removal efficiency of strain LCU1 treatment of perch aquaculture wastewater. (* indicates *p* value ≤ 0.05, *** indicates *p* value ≤ 0.001).

**Table 1 microorganisms-13-00297-t001:** Results of biochemical identification of isolated strains.

Reaction	LCU1	*Klebsiella pneumoniae*.
Colony size	Big and convex	Big and convex
Colony shape	Orbicular	Orbicular
Colony lustre	Smooth	Smooth
Colony edge	Neat	Neat
Colony colour	Pallid	Pallid
Colony transparency	Translucent	Translucent
Shape	Rod	Rod
Gram’s stain	−	−
Glucose fermentation	+	+
Lactose fermentation	+	+
Voges–Proskauer (V-P)	+	+
Ornithine decarboxylase	+	/
Hydrogen sulphide	−	/
Phenylalanine	−	/
Peptone water	−	/
Carbamide	+	+
Dulcitpl	+	d
Simmons citrate	+	/
Lysine	+	/

+ indicates positive; − indicates negative; / indicates no mention in the literature; d indicates that 11–89% of strains were positive.

## Data Availability

Data will be made available on request.

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
