# Peer review of "Characteristics and Mechanism of Ammonia Nitrogen Removal by Heterotrophic Nitrification Bacterium Klebsiella pneumoniae LCU1 and Its Application in Wastewater Treatment"

_microorganisms, 2025, doi:10.3390/microorganisms13020297_

Round 1
Reviewer 1 Report
Comments and Suggestions for Authors
The manuscript investigates the ammonia nitrogen removal potential of a newly identified Klebsiella pneumoniae strain (LCU1) for wastewater treatment. This study is significant due to the global challenge of nitrogen pollution and the need for efficient biological nitrogen removal processes. The authors comprehensively analyse strain LCU1’s nitrification capabilities under varying environmental conditions alongside its genome annotation. While the manuscript is well-structured and data-rich, certain aspects require improved clarity, scientific rigor, and presentation.
1. While the abstract summarizes the study effectively, it emphasize the practical implications and limitations of the findings more clearly.
2. Expand the rationale for focusing on Klebsiella pneumoniae LCU1, building on the background provided for heterotrophic nitrification.
3. Provide a more straightforward explanation for selecting experimental parameters and a more thorough discussion linking the results to existing literature.
4. Address the discrepancy between line 124 (“neighbor-joining method using MEGA 5.1 software”) and line 196 (“phylogenetic tree constructed in MEGA 11.0”). Ensure consistency.
5. Include detailed methods for genome sequencing, assembly, and annotation in the methodology section. Specify software versions and parameters used for these processes.
6. Upload the genome sequence to the NCBI genome portal and provide the accession number in the manuscript.
7. Ensure all axes in Figures 1 and 4 are labelled clearly and improve font sizes for better readability.
8. Maintain uniform formatting of units (e.g., mg/L, °C) throughout the manuscript.
9. Verify and ensure that genome size and gene count data are correctly reported in the results section.
10. Provide a clearer explanation for selecting experimental parameters and a more thorough discussion linking the results to existing literature.
11. The discussion should delve deeper into the implications of the genomic findings, particularly their connection to the observed nitrification efficiency.
12. Clarify the choice of specific heavy metal concentrations and salinity levels, particularly their relevance to typical aquaculture wastewater conditions.
13. Strengthen the connection between identified genes and the observed nitrification processes, providing clearer interpretations of genomic findings.
14. Quantitatively compare the performance of LCU1 with other nitrifying bacteria (e.g., Acinetobacter spp., Pseudomonas spp.) under similar conditions.
15. Analyze the genomic uniqueness of LCU1, particularly the absence of amo and hao genes, and discuss the potential implications for novel nitrification pathways.
16. Provide a detailed discussion on the pathogenicity and antibiotic resistance of Klebsiella pneumoniae LCU1, given its isolation from chicken fecal samples and its potential as an opportunistic pathogen. Discuss the safety implications for its use in bioremediation.
17. Increase the font size of figure labels and provide more detailed figure legends to ensure sufficient explanation of the data presented.
Author Response
-
Response 1: Thank you very much for your comments.
We have added a brief description of the shortcomings at the nitrogen metabolism pathway in revised version, as well as a concise overview of the current strain's limitations in actual wastewater treatment in revised version.
Revised version line 24: The genome analysis of strain LCU1 showed that key genes involved in nitrogen metabolism, including narGHI, nirB, nxrAB, and nasAB, were successfully annotated; hao and amo were absent, but the nitrogen balance analysis determined that the strain had a heterotrophic nitrification ability.
Revised version line 29: This suggests that with intensive colonisation treatment, the strain has promising application potential in real wastewater treatment.
-
Response 2: Thank you very much for your comments.
We already added the addition of information about the nitrogen metabolism pathways of other Klebsiella pneumoniae strains, as well as the denitrification capacity of Klebsiella pneumoniae, in revised version of the Heterotrophic Nitrification Background section of the Preface.
Revised version line 60 : HNB has a very good application effect in real wastewater treatment; the removal of NH4+-N by Alcaligenes faecalis strain No.4 is 5-40 times higher than that of microorganisms under the same conditions [13]. In studies of the simultaneous degradation of organic matter in wastewater by HNB, it was found that heterotrophic nitrification could explain the achievement of complete nitrification in systems with low dissolved oxygen and high organic loads [14].
Revised version line 77: The nor, nis, and nas were successfully annotated in the whole genome of Klebsiella pneumoniae strain EGD-HP19-C; although it was shown to be heterotrophically nitrifying, the nitrogen metabolism pathway was not fully described [17].
Revised version line 82: Klebsiella pneumoniae are widespread in nature. Fang et al. screened seven strains of Klebsiella pneumoniae in sludge; these strains also proved to be heterotrophically nitrifying, and the NH4+-N removal rate could be as high as 97.38% [19].
-
Response 3: Thank you very much for your comments.
We provide an explanation for why we chose to set these experimental parameters and discuss the existing results in comparison to the previous studies.
Revised version line 153: The six carbon sources selected in this study included common organic carbon sources, such as sugars, organic acids, and small molecular organic alcohols.
Revised version line 554: The appropriate C/N ratio is essential to maintain the normal functioning of microbial cells, while the ratio of carbon to nitrogen ratio also affects the growth of the strain and the ammonia nitrogen removal efficiency. Therefore, the effect of the C/N ratio on strain LCU1 is consistent with this inference, and its optimal C/N ratio is 10.In studies of the optimal C/N ratio, o ther nitrifying bacteria strains showed similar results, such as Pseudomonas stutzeri YG-24 with an optimal C/N ratio of 8:1 [26] and Agrobacterium sp.
-
Response 4: Thank you very much for your comments.
We have taken steps to ensure that the information provided about the software used is consistent and accurate.
Revised version line 136: The phylogenetic tree was constructed via the neighbour-joining method using MEGA 11.0 software.
-
Response 5: Thank you very much for your comments.
We have added detailed methods for sequencing, assembling, and annotating genomes in Material Methods in revised version .
Revised version line 176:Whole genome sequencing of strain LCU1 was conducted using a combination of whole genome sequencing technology from Majorbio (Shanghai, China) and Illumina and PacBio sequencing technologies. The resulting sequencing data exhibited a coverage of 124x, and 72,040 high-quality reads were reassembled using the SOAPdenovo (2.04) genomics workbench. Glimmer was employed for the purpose of CDS prediction, with the relevant annotations being obtained from the NR, Swiss-Prot, and KEGG databases. This was achieved by means of the utilisation of sequence alignment tools such as BLAST, Diamond, and HMMER. In summary, each set of query proteins was aligned with the aforementioned databases, and annotations of the best-matched subjects (e-value < 10-5) were obtained for gene annotation.
-
Response 6: Thank you very much for your comments.
We have taken the step of uploading the full genome information of the strain to NBCI.
Revised version line 425:The genome size was 5,751,502 bp (including scaffolds). Additionally, the repeat sequence was 20,782 bp, constituting 0.41% of the total length. The entire genome of LCU1 was uploaded to the NCBI (National Center for Biotechnology Information) database (accession: PRJNA1199685) and mapped based on the sequencing (Figure 5).
-
Response 7: Thank you very much for your comments.
We have taken care to ensure that each axis of Figures 1 and 4 is clearly labelled and adjusted to a suitable size for reading, with a view to ensuring legibility.
-
Response 8: Thank you very much for your comments.
We have adjusted the unit formatting in revised version to make it consistent with the full text.
Revised version line 115 : Chicken faecal samples were obtained from Woneng Agricultural Technology Co. Liaocheng City, Shandong Province, China, and returned to the laboratory at 4℃.
Revised version line 118 : Following thorough mixing, faecal samples (10 g) were inoculated in 300-mL triangular conical flasks containing 90 mL of sterilised enrichment medium, shaken thoroughly, and reincubated at 180 rpm and 30℃.
Revised version line 122 : The enriched bacterial suspension was diluted to 10-7 in a gradient, and 0.1 mL of a bacterial solution diluted to 10-5, 10-6,and 10-7 was applied to the plate screening medium and incubated in an inverted incubator for 2~3 days at 30℃.
Revised version line 126 : Single colonies were picked from the purified plates and incubated in 300-mL triangular flasks containing 100 mL of NH4+-N at 180 rpm and 30℃ for 32 h.
Revised version line 129 : The isolates with the highest NH4+-N removal activity were kept in slants of LB agar medium at 4℃.
Revised version line 568 : Most HNB strains, such as Pseudomonas stutzeri C3, are reportedly mesophilic, exhibiting optimal growth at temperatures between 30℃ and 35℃
-
Response 9: Thank you very much for your comments.
We rechecked the genome size data in revised version at results 3.5 section to ensure they were correct.
Revised version line 425 : The genome size was 5,751,502 bp (including scaffolds). Additionally, the repeat sequence was 20,782 bp, constituting 0.41% of the total length. The entire genome of LCU1 was uploaded to the NCBI (National Center for Biotechnology Information) database (accession: PRJNA1199685) and mapped based on the sequencing (Figure 5). The LCU1 genome comprises a circular chromosome and two plasmids. The prediction analysis showed that there were 5,342 coding genes in the gene sequences, with the total length of the predicted coding genes being 501,9144 bp. The gene average length of the sequence was 939.56 bp, with a GC content of 58.48 %. Furthermore, there were 25 rRNAs (comprising 8 23S rRNAs, 8 16S rRNAs,and 9 5S rRNAs) and 86 tRNAs. A summary of the characteristics of the assigned genes is provided in the Supplementary Information. Notably, 5,338 and 4,728 genes were annotated through the NR and Swiss-Prot databases, respectively. The genes associated with the nitrogen metabolism pathway were studied using the KEGG (Kyoto Encyclopaedia of Genes and Genomes)database . The strain demonstrated HNB functionality based on preliminary experimental analysis; however, KAAS (KEGG Automatic Annotation Server) was able to reveal the key genes involved in nitrification.
-
Response 10 : Thank you very much for your comments.
We provide an explanation for why we chose to set these experimental parameters and discuss the existing results in comparison to the previous studies.
Revised version line 153: The six carbon sources selected in this study included common organic carbon sources, such as sugars, organic acids, and small molecular organic alcohols.
Revised version line 554: The appropriate C/N ratio is essential to maintain the normal functioning of microbial cells, while the ratio of carbon to nitrogen ratio also affects the growth of the strain and the ammonia nitrogen removal efficiency. Therefore, the effect of the C/N ratio on strain LCU1 is consistent with this inference, and its optimal C/N ratio is 10. In studies of the optimal C/N ratio, other nitrifying bacteria strains showed similar results, such as Pseudomonas stutzeri YG-24 with an optimal C/N ratio of 8:1 [26] and Agrobacterium sp.
-
Response 11: Thank you very much for your comments.
We further related the genome sequencing results to the nitrification capacity of the LCU1.
Revised version line 612 : Regarding the annotation of the heterotrophic nitrification efficiency and nitrogen metabolic pathway of strain LCU1, this study showed that the strain could not directly utilise NO3--N and NO2--N when KNO3 and NANO2 were used as the sole nitrogen sources. However, strain LCU1 exhibited normal growth. The annotation results revealed the presence of nif in strain LCU1, which was postulated to facilitate nitrogen conversion by LCU1 through the action of nitrogen fixation enzymes. The same nitrogen fixation genes were detected in the genus Klebsiella michiganensis [37]. The key genes in the metabolism of heterotrophic nitrifying bacteria are amo and hao, but neither of these genes was annotated in LCU1 ; thus, it was speculated that genes with similar effects may exist [38]. When NH4+-N was the only nitrogen source, NO3--N accumulation was detected, and it was speculated that NH4+-N was converted to hydroxylamine in the process and that hydroxylamine was converted to NO2--N through a pathway and was oxidised to NO3--N by NXR, causing NO3--N accumulation. This nitrification process was rapid, as no NO2--N accumulation was detected, and a similar pathway was postulated in the study of SND-01 [39].
-
Response 12: Thank you very much for your comments.
Surplus fish feed deposition is also one of the sources of pollution of aquaculture wastewater, the content of Cu2+ in fish feed samples in North China was 8.84-33.24 mg/kg, and the content of Zn2+ was 67-215 mg/kg, so these two heavy metal ions were selected as the object of study. Shandong Province is a large province of metallurgy and chemical industry in China. When a large amount of manganese-containing and iron-containing residues from the process of manganese electrolysis are washed by rainwater in the process of stockpiling, free manganese and ammonia-nitrogen and other pollutants may enter into the neighbouring soils and waterbodies through surface runoff and underground seepage, which will be absorbed by the organisms in excessive amounts and cause manganese toxicity, therefore, manganese ions were used as the objects of the study.
The salinity of aquaculture wastewater is around 3%, the reason why 0-50% is chosen in this paper is to verify whether LCU1 can adapt to the more severe high salt environment, if it can still maintain a certain ability to remove ammonia nitrogen under high salt environment indicates that the strain is well adapted to the salinity, and it has the potential of treating high salt wastewater.
-
Response 13: Thank you very much for your comments.
We have taken your suggestions on board and made some additions to this section.
-
Response 14: Thank you very much for your comments.
The applicationthe ammonia nitrogen removal by Pseudomonas mendocina S16 immobilisation with sodium alginate, was 79.80% after 72h , whereas LCU1 was able to reach 83.70% within 24h in aquaculture wastewater treatment. The ammoniacal nitrogen removal rate of Acinetobacter sp TSH1 at low initial ammoniacal nitrogen concentration could reach 96.6% at 36h, while LCU1 could reach 93.30% at 24h. Therefore, we hypothesise that LCU1 has even better performance in removing ammoniacal nitrogen.
-
Response 15: Thank you very much for your comments.
We have analyzed the unique nitrogen metabolism pathway of LCU1 and discussed its possible causes.
Revised version line 626: The unique nitrogen metabolism mechanism of LCU1 can effectively remove NH4+-N but lacks the ammonia oxidation of the traditional nitrification pathway of ammonia oxidation genes. For example, Pseudomonas putida Y-9 can oxidise NH4+-N to nitric oxide and can further oxidise it to nitrous oxide under aerobic conditions, with no accumulation of hydroxylamine as an intermediate in the whole process and without the presence of the amo in the genome of its strain [40]. Another speculation is that there may be alternatives or redundant genes of amo and hao genes in LCU1, which may have similar functions but may differ in their sequences from the known genes; therefore, they cannot be accurately identified using the existing annotation algorithms. It is also possible that the gene annotation is based on existing databases and comparison algorithms; however, for some of the newly discovered strains or strains with special metabolic pathways, the existing databases may not be able to accurately annotate all their genes.
-
Response 16: Thank you very much for your comments.
When we apply Klebsiella pneumoniae LCU1 for nitrogen removal from aquaculture wastewater, the bacteria are not added directly to the aquaculture environment, but rather the wastewater is collected and then treated in a uniform manner. In addition to this, previous studies have screened for non-toxic Klebsiella pneumoniae (DZYC02).
Revised version line 644 : LCU1 is a Klebsiella pneumoniae strain isolated from chicken faeces. Hence, in the process of practical application, its direct use in the breeding environment should be avoided, and unprotected direct contact between operators and strain LCU1 should also be avoided in order to reduce its pathogenic potential. The resistance of Klebsiella pneumoniae to antibiotics varies among different strains, and although previous studies have shown that strain TN-1 can degrade tylosin [18], strain LCU1 has a very low ammonia nitrogen removal rate in aquaculture wastewater in which antibiotics have not been inactivated, possibly because LCU1 is sensitive to antibiotics.
-
Response 17: Thank you very much for your comments.We increased the font size of figure labels and provide more detailed figure legends.
-
We have made additional revisions to address other shortcomings in the article, in addition to the questions and suggestions you have already raised. Thank you for your valuable input.

Reviewer 2 Report
Comments and Suggestions for Authors
This article deals with the characterization of a new K. pneumoniae strain with nitrification activity. However, I must recommend its rejection due to two main reasons:
1) The document is challenging to read. It needs a profound English edition as many grammar errors are found. Moreover, some parts were copy-pasted from other articles, leaving traces of row numbers within words. Terminology, in several cases, is incorrectly used. I strongly recommend a revision by a professional proofreading service with experience in microbiology and genomics. The data is poorly presented (like using average and standard deviation instead of box or violin plots with individual data points and statistical significance)
2) The overall objective of the work is confusing. There is no clear hypothesis-guided discourse. The dogmatic presentation of ideas fully committed to the strain's practical application results in the disordered presentation of methods and results, omitting critical sections like the genomic analysis (which should be a separate work).
Comments on the Quality of English Language
Please make a professional proofreading service revise this work.
Author Response
-
Response 1:Thank you very much for your comments.
We have used a professional proofreading service to check the grammar and terminology of this article. During the proofreading process, we did not find any instances of line numbers being inserted, but we have adjusted the article accordingly for excessive repetition. All experimental designs were set up with three sets of replications, none of the data were presented as single point data, and the image size of the article was adjusted to ensure that the error lines were clearly presented.
-
Response 2: Thank you very much for your comments.
We isolated a strain of Klebsiella pneumoniae LCU1 and applied it to the removal of ammoniacal nitrogen, explored its optimal conditions for ammoniacal nitrogen removal and the mechanism of nitrogen removal, and applied it to aquaculture wastewater. We have elevated the lead-in to the abstract and added a preface about why we used Klebsiella pneumoniae denitrification in order to highlight the main idea of the paper. For the genomic data we provide a more detailed description, complemented by analysis methods and software in line 176.
Revised version :
2.6 Genome sequencing and gene annotation
Whole genome sequencing of strain LCU1 was conducted using a combination of whole genome sequencing technology from Majorbio (Shanghai, China) and Illumina and PacBio sequencing technologies. The resulting sequencing data exhibited a coverage of 124x, and 72,040 high-quality reads were reassembled using the SOAPdenovo (2.04) genomics workbench. Glimmer was employed for the purpose of CDS prediction, with the relevant annotations being obtained from the NR, Swiss-Prot, and KEGG databases. This was achieved by means of the utilisation of sequence alignment tools such as BLAST, Diamond, and HMMER. In summary, each set of query proteins was aligned with the aforementioned databases, and annotations of the best-matched subjects (e-value < 10-5) were obtained for gene annotation.
-
We have made additional revisions to address other shortcomings in the article, in addition to the questions and suggestions you have already raised. Thank you for your valuable input.

Round 2
Reviewer 1 Report
Comments and Suggestions for Authors
The revised manuscript has addressed all previous concerns effectively, and the updates have improved the clarity and quality of the work. In its current form, the manuscript is suitable for publication, and I recommend it be accepted for publication.
Author Response
- Comment 1: The revised manuscript has addressed all previous concerns effectively, and the updates have improved the clarity and quality of the work. In its current form, the manuscript is suitable for publication, and I recommend it be accepted for publication.
- Resoponse 1: Thank you very much for your recognition and recommendation of our revised manuscript! Your comments have greatly enhanced the clarity and quality of the article and we are deeply grateful for your expert guidance.

Reviewer 2 Report
Comments and Suggestions for Authors
The authors have made the requested general corrections. Please attend the following pendant:
1) Please change the graphs A-G from Figure 4 and the graph from Figure 7 to box and whiskers, showing individual data determinations.
2) Please provide a high-resolution for Figure 6. Or best, please provide a vectorial image (*.svg or *.eps). This circus plot needs to be analyzed for readers in detail, which the current image does not permit.
3) Please provide a detailed procedure for genomic analysis. Naming the company is not enough. Please cite the web page with public access to the used pipeline or attach the company's document stating the detailed procedures. Please include the sequencing QC report data as supplementary material.
Author Response
-
Comments 1: [Please change the graphs A-G from Figure 4 and the graph from Figure 7 to box and whiskers, showing individual data determinations.]
Response 1: Thank you for pointing this out.We agree with this comment.
We therefore replaced all bar charts in Figures 4 (A-G) and 7 with box plots to ensure a clear presentation of individual data.
-
Comments 2: [Please provide a high-resolution for Figure 6. Or best, please provide a vectorial image (*.svg or *.eps). This circus plot needs to be analyzed for readers in detail, which the current image does not permit. ]
Response 2: Thank you for pointing this out.We agree with this comment.
We have replaced the *.png with the sharper *.svg .
-
Comments 3: [Please provide a detailed procedure for genomic analysis. Naming the company is not enough. Please cite the web page with public access to the used pipeline or attach the company's document stating the detailed procedures. Please include the sequencing QC report data as supplementary material.]
Response 3: Thank you for pointing this out.We agree with this comment.
We have cited the URL of the publicly accessible analytical platform in the revised version and uploaded the QC report in the supplementary materials.
Revised version line 179:
All of the analyses were performed using the online platform of Majorbio Cloud Platform (http://cloud.majorbio.com) from Shanghai Majorbio Bio-pharm Technology Co.,Ltd.
